# Evaluations of psychosocial cancer support services: A scoping review

**Solveigh P. Lingens**[ID]*, **Holger Schulz, Christiane Bleich**

Department of Medical Psychology, University Medical Centre Hamburg-Eppendorf, Hamburg, Germany

* s.lingens@uke.de

**Data Availability Statement:** All relevant data are within the paper and its Supporting information files.

**Funding:** This study was funded by the Hamburger Krebsgesellschaft e.V. (Cancer Society Hamburg).

## Abstract

### Background

A diagnosis of cancer leaves most patients with cancer and their relatives with an increased psychological burden. Throughout the course of the illness, social, occupational or legal changes may lead to psychological distress. Psychosocial cancer support services offer psychological, social and legal support. However, little is known about the effectiveness of psychosocial support services implemented in health care. Therefore, this scoping review aims to provide an overview of current literature evaluating out-patient psychosocial support services.

### Methods

Databases searched were PubMed, PsycINFO, PSYNDEX, PsycArticle, Medline, Web of Science, Google Scholar, Cochrane, and Embase. Two independent researchers conducted the systematic search. We included studies that were published in English and assessed at least one patient reported outcome measure. Studies that assessed psychotherapy, online support or telephone counselling were excluded. The review was reported according to PRISMA-ScR guidelines. A search of the databases identified 2104 articles. After excluding duplicates, screening titles, abstracts and full-texts, 12 studies matching the criteria were identified.

### Results

One study was an RCT, six were prospective with no control group and five studies were cross-sectional with one measurement point. The most common outcome measures across studies were well-being, concerns and satisfaction with the support services.

### Conclusion

While the included studies indicate some improvements to well-being for patients with cancer, the low number and lack of high quality of studies indicate these findings should be interpreted with caution. However, high-quality research on the effectiveness of psychosocial support services is needed to determine that the interventions are effective.

HS and CB received the funding. The funders had no role in study design, data collection and analysis, decision to publish, or preparation of the manuscript.

**Competing interests:** The authors have declared that no competing interests exist.

## Introduction

According to the World Health Organization's latest global cancer data, 18.1 million new cases of cancer and 9.6 million deaths were recorded in 2018 [1]. Worldwide, approximately 43.8 million people are currently living with cancer. Fortunately, the survival time has improved continuously in the last years, as treatment is improving [2]. It has been established that patients living with or surviving cancer have poorer quality of life and often feel the need for support beyond their medical treatment. Psychosocial concerns such as psychological, occupational and financial obstacles are perceived as distressing by patients with cancer [3]. About 30–60% of patients wish to be supported in their psychosocial concerns [4–6]. A number of different psychosocial interventions are generally available and potentially help to address the psychological and social concerns following a cancer diagnosis [7, 8]. Depending on the problems that need to be addressed psychosocial interventions for patients with cancer can generally be assigned to one of the three following areas: mental health services, social work services and spiritual care [8–10]. Psychotherapy (e.g., CBT) is a mental health service that is recommended when there is an indication of a psychiatric diagnosis (according to DSM or ICD diagnostic manuals) [7]. Social work services address psychosocial and practical problems resulting from a cancer diagnosis [7]. Common types of psychosocial support are supportive psychotherapy, grief and educational counselling and support groups [7]. Spiritual care may be relevant for patients having specific concerns regarding their religious believes [7].

There is some indication of the efficacy of psychosocial support for patients with cancer in improving the quality of life, decreasing levels of distress and the risk for developing depression or anxiety disorders [11–15]. The interventions tested for efficacy and summarized in the reviews are largely controlled and standardized to reduce the likelihood of experimental bias. However, the interventions often need to be adapted according to the diverse needs of patients with cancer when implemented in the health care setting [16, 17]. Hence, implemented psychosocial interventions may be flexible in duration, contents, delivery format (e.g., inpatient or outpatient setting and online, face-to-face or telephone) and may be administered by different health care staff (e.g., psychologists, nurses, trained volunteers) [8, 16]. The efficacy of psychosocial interventions investigated in an experimental study may not be transferable when implemented to real-world psychosocial support services, due to the different settings and contexts of the support services. Therefore evaluations of implemented psychosocial support in out-patient health care are important to assess their effectiveness with consideration to their ecological validity and quality. There is currently no evaluation of implemented psychosocial cancer support within the out-patient settings.

Therefore, the aim of this review is to illustrate the extent and nature of international studies evaluating implemented psychosocial cancer support services in the out-patient setting. For this review psychosocial cancer support services are defined as psychosocial services offered at an outpatient facility. The services are understood as routine interventions implemented in health care and not as stand-alone interventions customized for research purposes. This review focuses on face-to-face delivered support since overviews of telephone and online support has been reported elsewhere [18, 19]. The primary objective of this review is to assess the number of studies and their characteristics such as their origin, study designs, study population, type of service, outcome measures and key findings regarding the evaluation of psychosocial cancer support services. This scoping review aims to provide a first overview of the available evidence in the field and thus may serve as a precursor for further reviews [20].

## Materials and methods

The findings of this scoping review are reported according to the Preferred Reporting Items for Systematic Reviews and Meta-Analyses Extension for Scoping Reviews (PRISMA-ScR).

### Eligibility criteria

All articles included had to evaluate psychosocial support services (including supportive psychotherapy) for patients with cancer or their relatives implemented in health care. The evaluation had to be performed by the clients of the services. At least one patient reported outcome measure had to be included. We included published manuscripts available in English language. There was no limitation to the years of publication to allow a broader search. Only studies assessing support delivered face-to-face were considered because reviews have already been conducted to summarize telephone and online support interventions [18, 19]. Therefore, articles that evaluated telephone or online support were excluded. Studies evaluating psychotherapy aimed at treating mental disorders were excluded.

### Information sources and search strategy

The following databases were searched for literature between 4th of June and 7th of October 2020: Ovid (Medline, PubMed, PsycINFO, PSYNDEX, Medline, PsycArticle), Web of Science, Google Scholar, Cochrane and Embase. There were no restrictions to the years of the studies searched. All articles published until the date of the search were considered.

The search terms were "cancer", "oncology", "counselling", "center", "service", "support", "psychosocial", "care", "supportive psychotherapy" and "outpatient". A full electronic search strategy for PubMed was developed to include all possible terms used for psychosocial support in British English and American English (e. g. counselling centre vs. counseling center) (S1 Table). The search strategy was translated into the other databases and adapted to their specific search connotations.

### Data charting process and data items

Two independent reviewers conducted the search. Firstly, titles were screened followed by an exclusion of duplicates. Secondly, abstracts were screened. The full-text review was also conducted independently. Differences regarding the final selections of articles were resolved by discussing the choices until a consensus was reached between the two researchers.

To extract the data at full-text review, a data charting tool was developed including the following items:

- Information on the article: Authors, year

- Study Design: Information on study design (e. g. RCT, observational, cross-sectional etc.), number of measurement time point (MTPs) and time of assessment

- Sample characteristics: Total number of participants included in the analyses, percentage of females, types of cancer, other information (if available) on treatment state and diagnosis

- Type of services: Information on type and content of support evaluated (e. g. psychological and social counselling)

- Delivered by: Health care professionals or others administering the support

- Delivered at: Name of institution where service is implemented, city and country

- Instruments: List of instruments and measurements used to evaluate the service

• Key findings: Results of the analyses in respondents to the instruments used (including statement of all significant improvements, percentage, p-values)

## Quality assessment

In a final step, the quality of articles included in the scoping review was assessed using the Effective Public Health Practice Project (EPHPP) Quality Assessment Tool [21]. Two independent researchers assessed the quality of the studies. Where the ratings did not correspond, a third researcher was consulted. The EPHPP evaluates the studies according to selection bias, study design (RCTs are considered highest quality), confounders, blinding, data collection methods, withdrawals and dropouts, intervention integrity and analysis [21]. A study can still be rated "strong" if it did not apply an RCT design. However, qualitative studies are generally rated weaker even if they are well conducted.

## Synthesis of results

For articles that used the same instruments a meta-analysis would be the preferable choice. However, due to the heterogeneity of the setting and context of psychosocial support services, comparability is not guaranteed. Hence, a narrative synthesis was the reasonable alternative. The articles were grouped together according to their choice of design (RCT, observational prospective, cross-sectional). For the significant key findings, the effect sizes were calculated. For the RCT- design and pre/post design Cohen's d was calculated, where a correlation of 0.5 between groups was assumed if not reported otherwise [22]. Furthermore, since different instruments were used for the evaluation of services, the results were compared across studies within similar theoretical concepts e.g., satisfaction with the service. The global scores and details of the quality assessment are reported in Table 1.

# Results

## Selection of sources of evidence

The literature search retrieved 2104 articles. After the exclusion of 235 duplicates, titles were screened, and 1549 articles were excluded. After the remaining 320 article abstracts were screened, 36 articles were identified for a full-text analysis. After the full text review, 12 articles were included in the review as they met all inclusion criteria. The most common exclusion criterion in the full-text analysis was the lack of a patient-reported outcome measure. The final article selection was first compared amongst both independent reviewers. Subsequently, deviating articles were discussed and in joint agreement, either included or excluded (Fig 1).

## Characteristics of sources of evidence

The study characteristics and results are summarized in Table 2. All articles were published after 2001, where eight articles were published after 2010. One study chose an RCT design [23], six studies were cohort studies with pre- and post-measures [24–29] and five studies were cross-sectional with retrospective measures [30–34]. The study populations comprised patients with all types of cancer for eight articles [24–26, 28, 29, 31–33], three included only patients with breast cancer [27, 30, 34] and one solely patients with breast and colon cancer [23]. One article stated a sample size of < 100 participants [25]. Eight articles reported a sample size between 101–500 participants [23, 24, 26–28, 30–32]. Two articles included between 501–1000 participants [29, 34] and one study included 1930 participants [33]. In all articles, at least 60%

**Table 1. Quality assessment rating (EPHPP).**

| First Author (year) | A SELECTION BIAS | B STUDY DESIGN | C CONFOUNDERS | D BLINDING | E DATA COLLECTION | F DROP-OUTS | Global |
|---|---|---|---|---|---|---|---|
| Seers (2009) | 2 | 2 | 1 | **3** | **3** | 2 | WEAK |
| | Participants represent target population but no participation rate | Cohort (one group pre-post) | No differences between groups | researcher and participant aware of status and study | not described | Response rate less than 80% | |
| Polley (2016) | 2 | 2 | 1 | **3** | 2 | **3** | WEAK |
| | Participants represent target population but no participation rate | Cohort (one group pre-post) | Controlled for at least 80% of confounders | researcher and participant aware of status and study | Valid but reliability not described | Follow-up rate less than 60% | |
| Edgar (2003) | **3** | 1 | 1 | 2 | 1 | 2 | MODERATE |
| | Less than 60% of participation | RCT | Controlled for at least 80% of confounders | Participants are not aware of the research question | Data collection tools are valid and reliable | follow up rate is less than 80% | |
| Ernst (2014) | 1 | **3** | 1 | 2 | **3** | 2 | WEAK |
| | Participants likely to represent target population | Cross-sectional retrospective | No differences between groups | Not described | Validity and reliability not tested | follow up rate is less than 80% | |
| Harrington (2012) | 2 | 2 | 1 | 2 | 2 | **3** | MODERATE |
| | Participants represent target population but no participation rate | Cohort (one group pre-post) | No differences between groups | Not described | Valid but reliability not described | Follow-up rate less than 60% | |
| Götze (2016) | 2 | 2 | 1 | 2 | 1 | 2 | STRONG |
| | Participants represent target population but no participation rate | Cohort (one group pre-post) | Controlled for > 80% of relevant confounders | Not described | Data collection tools valid and reliable | follow up rate is less than 80% | |
| Goerling (2010) | **3** | 2 | 1 | 2 | 1 | **3** | WEAK |
| | less than 60% participation | Cohort (one group pre-post) | Controlled for > 80% of relevant confounders | Not described | Data collection tools valid and reliable | Follow-up rate less than 60% | |
| Frenkel (2010) | 1 | 2 | **3** | 2 | **3** | **3** | WEAK |
| | 80% participation, likely to represent target group | Cohort (one group pre-post) | Not described | Not described | Reliability and Validity not described | Follow-up rate less than 60% | |
| Boulton (2001) | 2 | **3** | 1 | 2 | **3** | **3** | WEAK |
| | Participants represent target population but no participation rate | Cross-sectional retrospective | No differences between groups | Not described | Validity and reliability not tested | No follow-up | |
| Blum (2006) | 2 | **3** | **3** | 2 | **3** | **3** | WEAK |
| | Participants represent target population but no participation rate | Cross-sectional retrospective | Not described | Not described | Reliability and Validity not described | No follow-up | |
| Amin (2011) | **3** | **3** | 2 | **3** | **3** | **3** | WEAK |
| | Less than 60% participation | Cross-sectional retrospective | Mentioned confounders but not how they are controlled for | Researcher and participant aware of status and study | Validity and reliability not tested | No follow-up | |
| Baker (2019) | 2 | **3** | **3** | **3** | **3** | **3** | WEAK |
| | Participants represent target population but no participation rate | Cross-sectional retrospective | Not described | Researcher and participant aware of status and study | Validity and reliability not tested | No follow-up | |

Global rating: 1 = STRONG (no WEAK score), 2 = MODERATE (one WEAK score), 3 = WEAK (at least 2 WEAK scores)

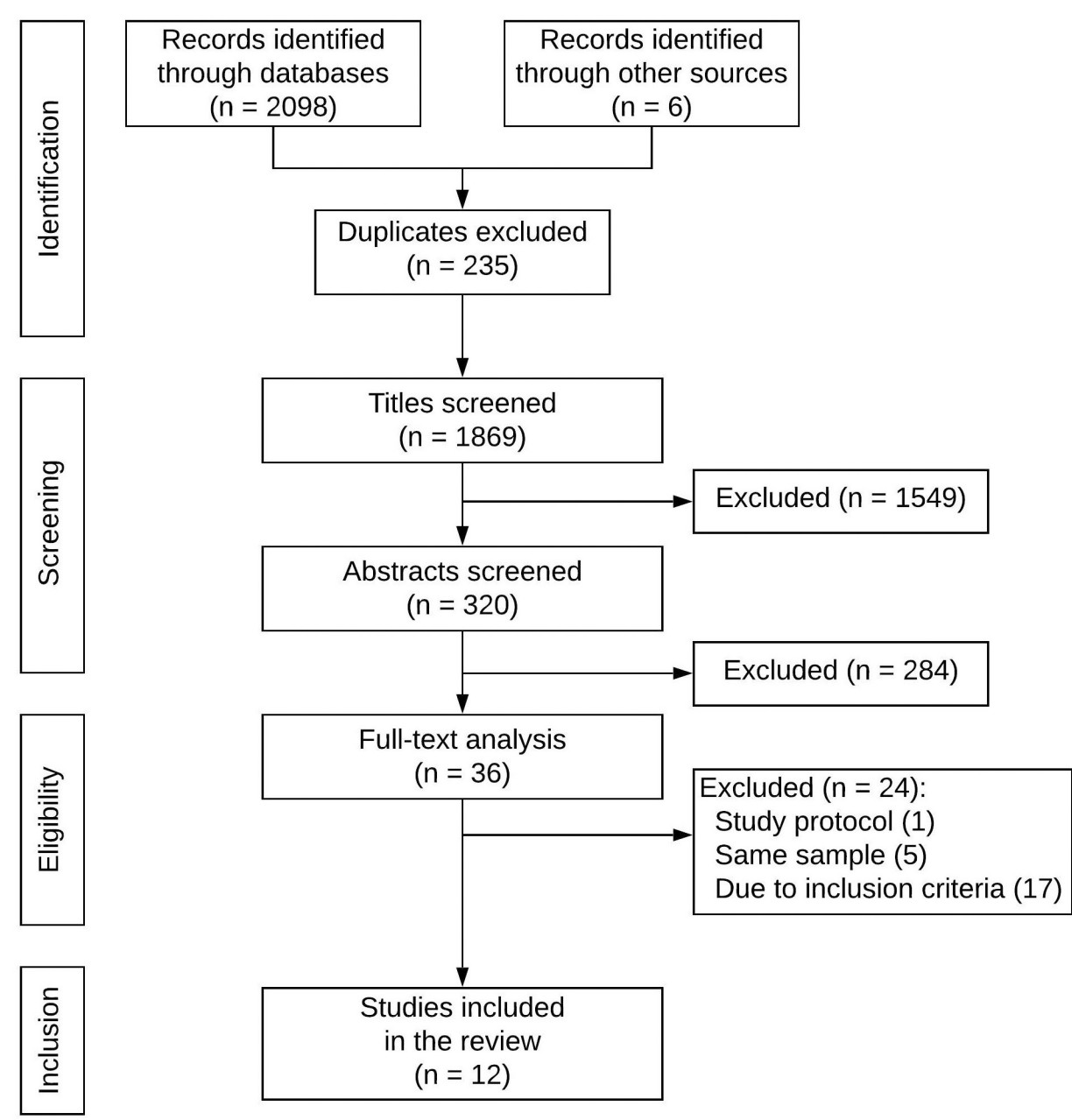

**Fig 1. Flowchart: Study screening and selection.**

of participants were female. The studies were conducted in four different countries, five in the UK [27–29, 32, 34], three in Germany [25, 26, 33], three in the USA [24, 30, 31] and one in Canada [23]. The type of service evaluated differ between the studies. In eight studies, the support services evaluated offered psychosocial support delivered by health care professionals with focus on psychological and/or social concerns [25, 26, 28, 29, 31–34]. Two studies focused solely on complementary medicine [24, 27] and two offered support by trained cancer survivors (volunteer mentoring) [23, 30]. Five articles collected data in three or more support centers [26, 27, 29, 33, 34].

**Table 2. Overview of selected articles (characteristics, services, instruments).**

| | Authors | Study Design | Sample characteristics | Type of service | Delivered by | Delivered at | Instruments | Key findings |
|---|---|---|---|---|---|---|---|---|
| Randomized Controlled Trial | | | | | | | | |
| 1 | Edgar, L. J. et al. (2003) | RCT with two arms 2 MTPs: —8 months after diagnosis —12 months after diagnosis | N = 177, (user: N = 138, non-user: N = 39), 86% female, breast and colon cancer, diagnosis within last 4 months | Various resources of support (Information, emotional support, Activities of daily living, Transportation, Finance/job)—average no. of resources used: 2.7 | Trained volunteers who had personal experience with cancer | Hope and Cope services, Montreal, Canada | POMS | No significant changes after 8 and 12 months |
| | | | | | | | FACT | Significant improvement of physical well-being after 12 months compared to non-users (p > .0018, d = 0.57) |
| | | | | | | | LOT | No significant changes after 8 and 12 months |
| Observational study with pre/post measure | | | | | | | | |
| 2 | Frenkel, M. et al. (2010) | 2 MTPs: Baseline— before first session—after 6–12 weeks | N = 238, 60% female, all types of cancer | One educational session on CIM use, CIM classes and therapies (Massage, acupuncture, music therapy, meditation, yoga, etc.) | Physician, staff nutritionist, other staff | University of Texas M D Anderson Cancer Centre's Integrative Medicine Clinic, Houston, USA | MYCaW | Leading concerns: 1. Information on complementary medicine 2. Physical problems; Severity of concerns decreased significantly after 6 weeks (p < .0001, d = 1.15), well-being increased significantly after 6 weeks (p < .0001, d = 0.41) |
| 3 | Goerling, U. et al. (2010) | 2 MTPs:— before first session—after at least 2 sessions | N = 46 baseline, N = 20 follow-up, 75% females, all types of cancer | Psycho-oncological and psychosocial counselling (Information, crisis intervention, basic psychotherapeutic services, social/legal issues, etc.)—2–5 individual sessions | Psycho-oncologists, social workers | Cancer counselling center, Berlin, Germany | FBK-R23 | Information deficit: significant improvement (p = 0.008)[2] Psychological distress: improvement but not significant (p = 0.08), social distress and everyday limitations: no improvement |
| 4 | Götze, H. et al. (2016) | 2 MTPs:—1 week after session—4 months after session | N = 213, 67.7% female, all types of cancer | Psycho-oncological and psychosocial counselling (Information, crisis intervention, basic psychotherapeutic services, social/legal issues, etc.) | Psycho-oncologists, social workers | Cancer counselling centers in Saxony (11), Germany | PHQ-9 | No significant improvement after 4 months |
| | | | | | | | GAD-7 | No significant improvement after 4 months |
| | | | | | | | SF-8 | No significant improvement after 4 months |
| | | | | | | | Distress Thermometer | Most common problem areas (57–77%): Fatigue, exhaustion, worry, anxiety, pain |
| | | | | | | | ZUF-8 | 90.8% of patients were satisfied with the counselling |

*(Continued)*

**Table 2.** (Continued)

| | Authors | Study Design | Sample characteristics | Type of service | Delivered by | Delivered at | Instruments | Key findings |
|---|---|---|---|---|---|---|---|---|
| 5 | Harrington, J. E. et al. (2012) | 2 MTPs:—before first session—after 6h/units of individual therapy | N = 402, 100% female, breast cancer | Nutrition, counselling, touch therapies, mind-body and medical herbalism, various groups and classes—6 sessions | Experienced specialist nurses, therapists | The Haven centres, London (1), Leeds (1), Hereford (1), UK | MYCaW | Leading concerns: 1. Psychological and emotional 2. Physical; significant improvement of concerns 1. (p < 0.0001, d = 1.86), 2. (p < 0.0001, d = 1.2) and well-being (p < 0.0001, d = 0.5) after treatment |
| 6 | Polley, M. J. et al. (2016) | 6 MTPs:—before course—after course—after 6 weeks—after 3 months—after 6 months—after 12 months | N = 135, 82.3% female, all types and stages of cancer (57% breast cancer) | Relaxation, meditation, mindfulness, and imagery, aimed at helping address physical, psychological, emotional, and spiritual health, as well as financial and relationship issues—2-day residential or a weekly non-residential course (2 hours over 7 weeks) | Medical doctors, nutritional therapists, and psychotherapists | Penny Brohn UK "Living Well with the Impact of Cancer" course, Bristol, UK | FACIT-SpEx | Significant improvement at 6 weeks after treatment: Emotional well-being (p ≤ .001, d = 0.29), functional well-being (p ≤ .05, d = 0.18), spiritual well-being (p ≤ .001, d = 0.26), overall score (p ≤ .001, d = 0.30), physical well-being improved after 6 months (p ≤ .01, d = 0.16), no significant improvement of social well-being |
| | | | | | | | MYCaW | Leading concerns: 1. Psychological and emotional 2. Concerns about well-being; Significant improvement of concerns 1. (p < 0.001, d = 1.06), 2. (p < 0.001, d = 1.00) and well-being (p < 0.001, d = 0.70) after treatment, still significant after 12 months |
| 7 | Seers, H. E. et al. (2009) | 2 MTPs:—before first session—4 weeks after end of course | N = 588, 91.7% female, all types and stages of cancer (79.2% breast cancer) | Psychosocial support, information and complementary therapy—12 h of complementary therapies | Medical doctors, nutritional therapists, and psychotherapists | The Haven centres, London (1), Hereford (1), Penny Brohn Cancer Care, Bristol, UK | MYCaW | Leading concerns: 1. Psychological and emotional 2. Concerns about well-being; Significant improvement of concerns 1. (p < 0.0005, d = 1.45), 2. (p < 0.0005, d = 1.23) and well-being (p < 0.0005, d = 0.49) after treatment |

Cross-sectional

(Continued)

**Table 2.** (Continued)

| | Authors | Study Design | Sample characteristics | Type of service | Delivered by | Delivered at | Instruments | Key findings |
|---|---|---|---|---|---|---|---|---|
| 8 | Amin, A. L. et al. (2011) | 1 MTP:—after counselling | N = 139, 100% female, breast cancer | Structured volunteer program: decrease sense of isolation, increase knowledge, coping strategies, provide hope | Trained survivors of cancer (volunteers) | After Breast Cancer Diagnosis (ABCD), Wisconsin, USA | Satisfaction | 96% would recommend the service "The ABCD program has helped me" was rated with a 5-point Likert mean rating of 4.41 |
| 9 | Baker, B. S. et al. (2019) | 1 MTP:—after counselling | N = 980, 98% female, breast cancer | Psychological support, help with treatment side effects and supported self-management activities—Up to 10 individual one-to-one sessions | Experienced BCH health care professional | Breast Cancer Haven (BCH) centres, London (1), Yorkshire (1) and Hereford (1), UK | Expectations, Concerns, Problems, Questions about service | 90.3% find the services much better than expected, physical concerns (67.2%), emotional concerns (77.6%) |
| 10 | Blum, D. et al. (2006) | 1 MTP:—after counselling | N = 243, 77% females, all types of cancer | Short-term psychosocial counselling—Up to 6 individual counselling sessions | Oncology social workers with mental health practices | Individual Cancer Assistance Network (ICAN), Florida, USA | CSQ-8 | 92% rated the intervention as being positive with 82% reported that they would return for intervention if needed |
| | | | | | | | Survey on concerns and goals | Main concerns were emotional support (80%) and meeting and negotiating the need of self and family (72%), the main goal was the increase of coping skills (51%), 74% found the fee service very important for the decision whether to seek intervention |
| 11 | Boulton, M. et al. (2001) | 1 MTP:—after one counselling session | N = 302, 78.1% female, all types of cancer | Humanistic counselling (facilitate self-knowledge, emotional acceptance and growth, personal resources)—Up to 8 individual counselling sessions | Accredited counsellors | British Association of Counselling (CancerBACUP), London, UK | Benefits Satisfaction | 90% felt emotional health was better after intervention, 95% were satisfied with the intervention |
| 12 | Ernst, J. et al. (2014) | 1 MTP:—10 weeks after first counselling session | N = 1930, 75.1% female, all cancer types, relatives included | Psycho-oncological and psychosocial counselling (Information, crisis intervention, basic psychotherapeutic services, social/legal issues, etc.)—at least one individual session | Psycho-oncologists, social workers | 26 cancer counselling centers, Germany | Satisfaction | Main concern was psychological support (relatives 86%, patients 68%), satisfaction with the center 81%, satisfaction with the intervention 76–78% |

MTP, Measurement time points; CIM, Complementary and Integrative Medicine, POMS, Profile of Mood States; FACT, Functional Assessment of Cancer Therapy; LOT, Life Orientation Test; MYCaW, Measure Yourself Concerns and Wellbeing; FBK-R23, Questionnaire of cancer patient distress—revised (German: Fragebogen zur Belastung von Krebskranken); PHQ-9, Patient Health Questionnaire; GAD-7, Generalized Anxiety Disorder; SF-8, Short Form-8 Health Survey; ZUF-8, Patient satisfaction (German: Patientenzufriedenheit); FACIT-SpEx, Functional Assessment of Chronic Illness Therapy Questionnaire with an additional spiritual subscale; CSQ-8, Client Satisfaction Questionnaire

## Results of individual sources of evidence

**Measurements and instruments.** The most common instrument used was the Measure Yourself Concerns and Wellbeing questionnaire (MYCaW) [35], which four studies chose as a primary outcome [24, 27–29]. The questionnaire was developed in the UK to evaluate cancer support services. It assesses patients' two main concerns for counselling with open-ended questions and their well-being with one item. The validity has been established; however, there are no indications for reliability [35]. Eight of the articles used at least one standardized outcome measure with acceptable psychometric properties [23–29, 31]. The standardized instruments included assessed psychological outcomes such as quality of life, distress, well-being, anxiety, or depressive symptoms. Five articles included measures on patient satisfaction [26, 30–33] and six articles assessed patients' concerns [24, 27–29, 31, 34].

**Key findings.** The results of the RCT showed that physical well-being improved after the supportive intervention compared to the control group but psychological and social well-being did not [23, 36]. All four pre- and post- studies using the MYCaW demonstrated improvement in general well-being [24, 27–29]. Polley et al. measured different aspects of quality of life with the Functional Assessment of Chronic Illness Therapy Questionnaire with an additional spiritual subscale (FACIT-SpEx) [37] and found a significant increase in emotional, functional and spiritual well-being after the support intervention [28]. Physical well-being improved six months after the support intervention [28]. The most common reported concerns of patients were emotional and psychological concerns and concerns about well-being [27–29, 31, 33]. One study identified information need on complementary medicine and physical problems as main concerns [24]. All articles that examined concerns remarked a significant decrease in the severity of concerns [24, 27–29]. One study found a significant improvement in information deficit measured by the German revised questionnaire about distress of cancer patients (Fragebogen zur Belastung von Krebskranken—revised; FBK-R23) [25, 38]. Across all cross-sectional studies at least 75% of patients indicated they were satisfied with the support services [26, 30–33]. The most common concerns were emotional and physical concerns and the wish to negotiate the need of self and family and improve coping skills [31, 34].

## Discussion

The aim of this scoping review was to identify the extent of research conducted, which evaluates psychosocial cancer support services. The characteristics and results of the included articles were assessed. To the authors' knowledge, this is the first literature review focusing on studies that assessed psychosocial support services for patients with cancer implemented in out-patient health care. Primarily, our results reveal the lack of high-quality research on the evaluation of psychosocial cancer support services—specifically in form of RCTs. The lack of studies with RCT designs may be due to the lack of standardized interventions offered in psychosocial support, which may hamper comparability. Another reason may be the difficulty of integrating a control group when assessing an intervention already implemented in health care, since it would be unethical to deny support for patients in need. However, other forms of psychosocial support for patient groups such as infertile women and men or patients with psychological problems have been successfully assessed in several RCTs [39, 40]. Hence, comparable studies are available and may be used as models for research on psychosocial support services for patients with cancer. One possible study design to overcome this concern could involve using wait-list controls.

The quality assessment of the included articles resulted in a "weak" rating for nine articles. This is largely due to the cross-sectional study design and the absence of standardized

instruments. The study of Götze and colleagues, which was rated as "strong" demonstrates that it is possible to conduct high-quality research in the field [26]. The quality assessment suggests that research on the effectiveness of implemented support services in cancer needs to be improved.

The characteristics of the selected studies suggest that evaluations of psychosocial support services might be more prevalent in Europe and Northern America, include more women than men, and most of the services evaluated are accessible for people with any type of cancer diagnosis. The reason the selected studies originate from Western countries may be due to the inclusion criteria of this review, as we only included research in English language. Another reason may be that those countries have more resources available for research on supportive services. On the contrary, psychosocial support services may be more common in those countries which in turn may result in a greater need for research. For example, one study investigated disparities in providing psychosocial care in 28 countries and found an underdevelopment of implemented psychosocial services mainly in Asia and Africa [41]. The study identified as the main reason for the insufficient psychosocial support offers in cancer care the lack of funding of health insurances and public health institutions [41]. Greater participation of women than men in the evaluated studies may be due to increased willingness of women to participate in research studies. However, it could also be that more women than men make use of support services. This would be consistent with studies on gender differences in attitudes towards help-seeking and desires for psychosocial support, where women show a greater desire for and more positive attitudes towards support compared to men [42, 43]. Two support services targeted specifically breast cancer patients and ten support services offered support for patients with any cancer diagnosis, which is also in line with the findings of a study assessing the characteristics of patients in multiple cancer support centers [44]. The heterogeneity of types of psychosocial support services offered may be due to the different standards and health care systems of each country [41]. In Germany, for instance, most psychosocial cancer support centers are subordinated to the German Cancer Society and thus the services are similar across the country [45]. In the UK, some cancer support services are free of charge and funded through the NHS, but most support services are being offered by private funding sources and thus the services offered may differ. Nevertheless, most psychosocial support services offer educational, emotional, and psychological support and some services offer social support or information on complementary medicine.

The heterogeneity of the services and their offers makes it difficult to interpret and compare the outcomes across the selected studies. The instruments used for the evaluation of the services differ across outcome measures as well as their psychometric properties. The results of the studies included in this review can be divided into three outcome groups: Psychosocial outcome measures, and the assessment of concerns and of the satisfaction with the services.

*Psychosocial outcome measures* were applied in seven studies. In psychosocial research, it is common to assess different outcome measures, e. g. anxiety, distress, depression, quality of life, because different health aspects may be addressed [12]. Most of these outcome measures had acceptable psychometric properties. However, only different aspects of well-being showed significant improvements. General and psychological well-being measured by the MYCaW and FACIT-SpEx resulted in clinically significant improvement [24, 27–29]. The improvement of well-being when measured by the MYCaW may be due to the proximity of its assessment with that of the concerns (Concerns are examined followed by well-being). The patient may connect the successful resolution of the concerns with an improvement of well-being. Further research is needed to study the improvement of well-being when assessed in combination with concerns in the setting of psychosocial support. Physical well-being measured by the FACT and FACIT-SpEx also improved after the support intervention [23,

28]. Although in one study, the first significant improvement of physical well-being was after six months [28]. Since this study did not compare the results to a control group, an inference from the relationship between intervention and physical well-being is limited. The lack of significant changes of the other instruments may be due to the insensibility of the instruments to detect small effects that psychosocial support may have on the patients. Moreover, patients could have experienced short-term mental health improvements, which may have diminished long-term. Studies with more measurement time points and instruments that are more sensible for short-term changes (i.e., the distress thermometer) may yield greater effects in psychosocial research [7, 46].

*Concerns* were assessed by self-developed items but mostly by the MYCaW, which is a partly standardized tool developed to evaluate psychosocial cancer support services [35]. The first part of the questionnaire captures the two main concerns of the clients before the session by means of open questions. The second part is applied after the session and assesses with a number of validated items the extent to which these concerns were successfully addressed and resolved. The most common concerns stated were of psychological nature, which is in line with the research on needs of oncological patients [47]. The findings imply that an assessment of concerns may improve the understanding of needs of patients with cancer and allow a more precise improvement of the services offered.

The *satisfaction* with the services was predominantly evaluated by qualitative measures or single and self-developed items except for the German version of the Client Satisfaction Questionnaire (Patientenzufriedenheit; ZUF-8) [26, 48]. Across all studies, the satisfaction was rated relatively high, which is likely to be due to an acquiescence bias [49, 50]. This bias is often observed when assessing satisfaction, as participants tend to agree with a positively framed item. The lack of standardized instruments may limit the quality of the evaluation and thus the informative value on the quality of the service. Instead of measuring satisfaction with the services, it may be more informative to focus on health-related outcomes or whether the concerns were sufficiently addressed.

## Limitations

By evaluating the findings of this scoping review, several limitations have to be taken into consideration. One limitation may result from the literature search. Since there is no standardized designation for psychosocial cancer support services, there is a possibility that some terms may not have been included. A further restriction could have resulted from the variety of combinations of the different designations. Hence, it could have been possible to combine "cancer" or "psychosocial" with "counselling" or "support" and with "service" or "center" or none. Furthermore, similar terms like "counselling" may have different meanings in different languages and countries. In Germany, psychotherapy would not be referred to as counselling whereas in English speaking countries, the term "counselling" is also used in the context of therapy. Nevertheless, the article titles and abstracts were screened thoroughly and conservatively to avoid false exclusions.

Another limitation is that only articles published in the English language were included in the review. Therefore, articles assessing the topic and written in other languages may have been missed in the search. Moreover, many more countries (e.g., Australia, India) provide psychosocial cancer support services but simply lack research on evaluating their services.

Despite these limitations, this scoping review provides a first overview of existing research on evaluations of psychosocial cancer support services, which highlights the lacking evidence on this important health care service.

## Conclusions

Overall, the results of this scoping review suggest that the current state of research on the effectiveness of psychosocial cancer support centers is unsatisfactory especially as high-quality studies including an RCT-design and standardized instruments assessing the full scope of the support services have not yet been conducted. Nevertheless, the results of this review imply that psychosocial cancer support services improve well-being and successfully address distressing psychosocial concerns of cancer patients and their relatives. Cancer support services have the ability to reach a higher number of patients with cancer faster and thus gain importance in cancer care. Therefore, evaluations of the effectiveness of the services are needed to maintain and optimize the quality of the support offered and to strengthen the evidence, that psychosocial support services are meaningful and relevant health care offers for cancer patients. Convincing evidence indicating that implemented psychosocial support services improve outcomes for patients with cancer may then result in increased funding. Consequently, an increase in funding would contribute to a growth and spread of the services to more countries and regions to reach and help a larger number of patients with cancer and their relatives.

## Supporting information

**S1 Table. Full search history (PubMed).**
(DOCX)

**S2 Table. PRISMA-ScR checklist.**
(DOCX)

## Acknowledgments

We would like to thank Noemi Struckmeier for her time and effort she has invested as a second reviewer.

## Author Contributions

**Conceptualization:** Solveigh P. Lingens, Holger Schulz.

**Data curation:** Solveigh P. Lingens.

**Formal analysis:** Solveigh P. Lingens.

**Methodology:** Solveigh P. Lingens.

**Supervision:** Holger Schulz, Christiane Bleich.

**Validation:** Solveigh P. Lingens.

**Visualization:** Solveigh P. Lingens.

**Writing – original draft:** Solveigh P. Lingens.

**Writing – review & editing:** Solveigh P. Lingens, Christiane Bleich.

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
