## [Decision Letter · Decision Letter 0]

14 Aug 2020

PONE-D-20-17447

Evaluations of psychosocial cancer counselling services: A scoping review

PLOS ONE

Dear Dr. Lingens,

Thank you for submitting your manuscript to PLOS ONE. After careful consideration, we feel that it has merit but does not fully meet PLOS ONE’s publication criteria as it currently stands. Therefore, we invite you to submit a revised version of the manuscript that addresses the points raised during the review process.

We look forward to receiving your revised manuscript.

Kind regards,

Tim Luckett

Academic Editor

PLOS ONE

Reviewers' comments:

Reviewer's Responses to Questions

**Comments to the Author**

1. Is the manuscript technically sound, and do the data support the conclusions?

Reviewer #1: Partly

Reviewer #2: Partly

2. Has the statistical analysis been performed appropriately and rigorously? 

Reviewer #1: N/A

Reviewer #2: N/A

3. Have the authors made all data underlying the findings in their manuscript fully available?

Reviewer #1: Yes

Reviewer #2: Yes

4. Is the manuscript presented in an intelligible fashion and written in standard English?

Reviewer #1: Yes

Reviewer #2: Yes

5. Review Comments to the Author

Reviewer #1: This paper is a scoping review. It aims to provide an overview of the current literature assessing the effectiveness of face-to-face counselling intervention in improving cancer patients’ outcome. Overall, this paper has significant value in an area where research has been lacking, opening the way to meet the needs of cancer patients better. This paper is generally well-organised and clearly written. I have some comments that might help make this paper more informative.

Most importantly, the overlaps of the terms “counselling” and “supportive psychotherapy” have important implications that need to be addressed. The main claim of the authors was that the current literature on the effectiveness of cancer counselling service lacks high quality studies, thereby the need for further research. This claim is supported by the findings in the review using the search strategies exploring the interventions of counselling while excluding psychotherapies.

There is contention around the similarities and differences of the terms “counselling” and “supportive psychotherapy”. One might argue that these terms are synonymous (e.g. Lebow 2019 Overview of Psychotherapies. UpToDate), with “supportive psychotherapy” being the technical term of “counselling”. Meanwhile, others might argue that for logistical reasons, these two terms are distinctly different (e.g. trained counsellors can offer counselling but not any form of “psychotherapy”). Given the aim of the scoping review is to provide a comprehensive overview of the current literature of studies assessing the effectiveness of counselling services (presumably at the technical level), performing the literature search excluding all forms of psychotherapy might have excluded studies that involve counselling interventions in nature but were labelled as “supportive psychotherapy”. This needs to be discussed more in the manuscript. I would recommend searching for “supportive psychotherapy” as well in the search strategy, while maintaining the exclusion of other forms of psychotherapies. This would ensure that the entirety of the literature is assessed.

Other points by various sections include:

AIM/OBJECTIVE

In the aim / objective section (line 70 – 71), suggest reversing the words “aim” and “objective”. I.e. “The aim of this review…” “The primary objective of this scoping review…”

METHODS

Consider adding the information about the years of the studies searched. This would make this review more informative in the future when comparison is required (especially in light of the recent COVID-19 outbreak with many counselling services transitioning to using video-conferencing facilities).

For the eligibility criteria, did the included studies need to have patient reported outcomes (PRO) as suggested in the Result section on line 123? If so, the eligibility criteria need to reflect that, with the limitation discussed (as studies with only therapist reported outcome would be missed). Meanwhile, line 169 in the Results section suggested that there was a study that included no PRO but was solely reliant on the counsellor’s report? Please clarify this discrepancy.

Line 86 – PubMed, PsycINFO and other databases are placed in parenthesis for Medline. What do the authors imply? That similar search strategies were used? Or these databases are equivalent? Please clarify.

For the Data Charting Process and in the associated Table 2, consider including more detail on the frequency of the counselling intervention (e.g. single one-off vs weekly session) for each included study, as the frequency can potentially influence the effectiveness.

RESULTS

Line 135 – “Eight” studies were cross-sectional – but the table of included studies showed only seven studies. Please clarify.

Parts of the Outcome section of the Results do not have references for the included studies being referred to, making it hard to cross-check/follow. The provision of which would be informative (e.g. Line 155).

In the Outcomes section, the authors mentioned that “wellbeing only improved when measured by MYCaW” (line 163) but previously also stated that FACT showed significant improvement of physical wellbeing after counselling (line 160). Please clarify the discrepancy.

Line 166 – “the most common concerns of patients were emotional psychological concerns”. Can the authors provide more evidence to support the claim? (e.g. specifying in the tables of included studies the various types of concerns reported)

DISCUSSION

As mentioned, consider expanding on the terms of counselling and “supportive psychotherapy” further, with the implication of the decision to include /exclude “supportive psychotherapy” in the search strategies discussed.

CONCLUSION

The starting sentence in the Conclusion of the manuscript does not match the primary concluding remark in the Abstract. Perhaps consider restructuring the sentences in the Conclusion section of the manuscript so that the lack of high-quality research is highlighted at the start of the section (e.g. bringing the sentence “The current state of research…is unsatisfactory” [line 264] forward and place the sentence on the suggested effectiveness of cancer counselling [line 258] later in the section).

Sentences (line 260-263) justifying the significance of counselling services compared to psychotherapy might be best placed elsewhere in the manuscript (e.g. introduction or discussion) with elaboration on the similarities and differences between them.

Figure 1: Suggest adding caption.

Table 2: Overview of selected articles: 1. Consider adding in the quantification of significant improvement for particular study (e.g. >30% improvement) to allow for easier appreciation of the effect size. 2. Consider adding the frequency intervention as mentioned above

Table 3: Instruments and key finding: There are some overlaps of contents between Table 2 and Table 3. Perhaps, the overlapping contents can be integrated?

S1 Table: Full search strategy. Elaborating on how the various search items are integrated in the final search would allow reproducibility. (e.g. Are all #1-15 joined with “OR”?). Consider adding in “supportive psychotherapy” in the search as suggested earlier.

With revision, this study would likely be an important contribution to the current literature where evidence for counselling service in this context is lacking.

Reviewer #2: Thank you for the opportunity to review this study. The review seeks to examine the literature to determine the scope of studies examining cancer counselling services. The topic area is important and the manuscript, for the most part, is concise and well-constructed. Nevertheless, the language needs to be tightened throughout. There are several colloquialisms or poorly worded sentences in the manuscript. It could be greatly improved by some careful editing. Further suggestions for improving the manuscript are outlined below.

Intro

- Line 47-48: Perceived as distressing – is this for cancer patients? Please clarify.

- Could consider using person first language, i.e. ‘patients with cancer’ rather than ‘cancer patients’.

- The authors don’t address any previous reviews examining the area of psychosocial cancer care in the background section. There have certainly been reviews in this domain. It would be helpful if the authors addressed this and identified how their current study fits in and builds on this literature.

Methods

- Eligibility criteria: Was there any restriction on study types? Would be good to include study designs that were excluded. Further, a rationale behind why some of the services (psychotherapy, online etc.) were excluded would be helpful.

- Data charting process and data items: Were discussions to resolve just between the two coding authors or were additional authors consulted? Please clarify.

- Synthesis of results: Some of this information is not really needed here. Would consider removing the first two sentences and sentence four.

Results

- Characteristics: Would suggest trying be more explicit with all the outlined characteristics. For example, rather than ‘Most articles reported a sample size between xxx’, should state ‘XX articles reported a sample size <100, XX between 101-400 and XX >400’. Obviously with consideration to the sample size spread or informed by literature on appropriate sample size.

- Table 2: Not sure I understand why the journal needs to be included in this table. I would suggest including the lead author, year and country all in one column (the first column). Remove journal and perhaps move the study design to the second or third column to contextualise the study. It is indicated that for the Key findings the P-value would be included in the table but none are reported and most don’t include proportions.

- Outcomes: I don’t think the way the outcomes are presented is the optimal way to present these results. Grouping by outcome measures implies that it is the measure that has impacted the findings rather than the intervention/perceptions of patients. I would suggest presenting the outcomes grouped by studies instead.

- There is no discussion of the methodological coding applied and how the scoring was applied to each study.

Discussion

- Further to my comment in the introduction, I am not sure this is the only review on this topic. It would be good for the authors to consider reviews in this field to highlight the gap they are addressing.

- Line 177 – remove ‘great’. Need to be cautious using terms like this in scientific writing.

- Lines 188-191: Could you provide a bit more detail on the studies you refer to? Were these studies population level? Were they outside the scope of this review?

- Line 189: Most services evaluated in this review were suitable – not just ‘most services are suitable’. Current phrasing is too broad and can’t be stated with accuracy.

- The third paragraph of the discussion is quite long and convoluted. It is difficult to follow the authors’ arguments in sections. I would recommend trying to refine this paragraph and talk about the points more explicitly.

- Not sure I agree that the instrument variation indicates a lack of research. It would be unusual for there to be a tool specifically targeting a service and population group. Differing outcome measures across quality of life, depression, anxiety etc. is very common in psychosocial literature. Agree that there is scope to talk about standardised psychosocial instruments for oncology patients, but I think the way this is currently written could be misleading.

- Para 4 on methodology basically just talks about the coding tool used – there is no real discussion of quality here and implications for the field. This paragraph needs to be strengthened.

- The delineation between cancer counselling services and psychotherapy is not clear in the manuscript so it makes it hard to understand some of the final conclusions. Could consider adding some detail to this in the intro.

6. PLOS authors have the option to publish the peer review history of their article (what does this mean?). If published, this will include your full peer review and any attached files.

Reviewer #1: **Yes: **Wei Lee

Reviewer #2: No

---

## [Author Response · Author response to Decision Letter 0]

13 Jan 2021

Reviewer #1: 

This paper is a scoping review. It aims to provide an overview of the current literature assessing the effectiveness of face-to-face counselling intervention in improving cancer patients’ outcome. Overall, this paper has significant value in an area where research has been lacking, opening the way to meet the needs of cancer patients better. This paper is generally well-organised and clearly written. I have some comments that might help make this paper more informative.

Comment 1:

Most importantly, the overlaps of the terms “counselling” and “supportive psychotherapy” have important implications that need to be addressed. The main claim of the authors was that the current literature on the effectiveness of cancer counselling service lacks high quality studies, thereby the need for further research. This claim is supported by the findings in the review using the search strategies exploring the interventions of counselling while excluding psychotherapies.

There is contention around the similarities and differences of the terms “counselling” and “supportive psychotherapy”. One might argue that these terms are synonymous (e.g. Lebow 2019 Overview of Psychotherapies. UpToDate), with “supportive psychotherapy” being the technical term of “counselling”. Meanwhile, others might argue that for logistical reasons, these two terms are distinctly different (e.g. trained counsellors can offer counselling but not any form of “psychotherapy”). Given the aim of the scoping review is to provide a comprehensive overview of the current literature of studies assessing the effectiveness of counselling services (presumably at the technical level), performing the literature search excluding all forms of psychotherapy might have excluded studies that involve counselling interventions in nature but were labelled as “supportive psychotherapy”. This needs to be discussed more in the manuscript. I would recommend searching for “supportive psychotherapy” as well in the search strategy, while maintaining the exclusion of other forms of psychotherapies. This would ensure that the entirety of the literature is assessed.

Reply: Thank you very much for this valuable remark. We conducted further research on the terms and definitions, and we came to realize that there is indeed some overlap with the term “psychosocial counselling” and “supportive psychotherapy” in the literature. We have included an additional search with the suggested term and edited the manuscript where applicable. However, none of the articles assessed in the additional search met the inclusion criteria so we did not include any further articles. 

This review focuses on evaluations of psychosocial cancer support services, which was not clarified sufficiently in the original manuscript. Thanks to your and the other reviewers’ comments, we have realized that the manuscript lacked a clear definition, which we have addressed now and led to an improvement of the consistency of the whole manuscript. We defined psychosocial cancer support services for this review as “psychosocial services offered at an outpatient facility and as routine interventions implemented in health care and not as stand-alone interventions customized for research purposes.” (line 76-79). We strengthened the inclusion criteria accordingly and had to exclude two studies from the original manuscript. The study by Öhlén et al. (2005) had to be excluded because the service was offered in an in-patient setting. The second study, by Ihrig et al. (2019) was excluded because it evaluated the process of an evaluation of cancer support services but not the services itself. 

To refer to all types of interventions including supportive psychotherapy and counselling, we have decided to change the terminology throughout the whole manuscript including the title to “psychosocial cancer support services” to avoid any confusion. 

Other points by various sections include:

AIM/OBJECTIVE

In the aim / objective section (line 70 – 71), suggest reversing the words “aim” and “objective”. I.e. “The aim of this review…” “The primary objective of this scoping review…”

Reply: Thank you. The changes were made as suggested.

METHODS

Consider adding the information about the years of the studies searched. This would make this review more informative in the future when comparison is required (especially in light of the recent COVID-19 outbreak with many counselling services transitioning to using video-conferencing facilities).

Reply: We have included the years of the studies searched: “There were not limits to the years of the studies searched. All articles published until the date of the search were considered.” (line 94-95)

For the eligibility criteria, did the included studies need to have patient reported outcomes (PRO) as suggested in the Result section on line 123? If so, the eligibility criteria need to reflect that, with the limitation discussed (as studies with only therapist reported outcome would be missed). Meanwhile, line 169 in the Results section suggested that there was a study that included no PRO but was solely reliant on the counsellor’s report? Please clarify this discrepancy.

Reply: Thank you very much for this comment. We have in fact overlooked that article 15 has not used a patient reported outcome measure. Therefore, we have decided to exclude this article as it also does not add any additional value to the review and does not comply with our eligibility criteria as you mentioned. 

Line 86 – PubMed, PsycINFO and other databases are placed in parenthesis for Medline. What do the authors imply? That similar search strategies were used? Or these databases are equivalent? Please clarify.

Reply: We searched the Databases in parenthesis via Ovid. We edited the section accordingly: “Ovid (Medline, PubMed, PsycINFO, PSYNDEX, Medline, PsycArticle)”. (line 104)

For the Data Charting Process and in the associated Table 2, consider including more detail on the frequency of the counselling intervention (e.g. single one-off vs weekly session) for each included study, as the frequency can potentially influence the effectiveness.

Reply: We absolutely agree with you and believe the results will benefit from more detail about the support services. We have included more information on the support services offered by adding more columns to Table 2: Type of service (content, frequency of sessions), delivered by (e.g., health care professionals), delivered at (name and location of center or service).

RESULTS

Line 135 – “Eight” studies were cross-sectional – but the table of included studies showed only seven studies. Please clarify.

Reply: Thank you for the remark. With the changes we have made in the final study selection a total of “five” studies were-cross-sectional. 

Parts of the Outcome section of the Results do not have references for the included studies being referred to, making it hard to cross-check/follow. The provision of which would be informative (e.g. Line 155).

Reply: Thank you. We have included citations for all results. 

In the Outcomes section, the authors mentioned that “wellbeing only improved when measured by MYCaW” (line 163) but previously also stated that FACT showed significant improvement of physical wellbeing after counselling (line 160). Please clarify the discrepancy.

Reply: Thank you for pointing out this alleged contradiction. The MYCaW measures general well-being whereas the FACT found that only physical well-being improved significantly. We added some more detail to the section on key findings and restructured the paragraph as result of the other reviewers’ suggestions. Please refer to line 177-186.

Line 166 – “the most common concerns of patients were emotional psychological concerns”. Can the authors provide more evidence to support the claim? (e.g., specifying in the tables of included studies the various types of concerns reported)

Reply: Thank you for this remark. We had two reasons why we only included the most common concerns reported in the articles. First, some articles only reported the most common concerns because most of the time they were measured by the MYCaW, which asked for the two most important concerns as open questions. Other studies assessing concerns also used open questions. Secondly, listing all concerns (if reported in the article) would go beyond the scope of the tables and this review. Generally, we decided to only include the relevant key findings of the studies, since the aim of this review was to provide a first overview of the studies on the topic. A few examples from the table 3 are: 

“Harrington, 2012: 1. Psychological and emotional 2. Physical; significant improvement of concerns 1. (p < 0.0001, d = 1.86), 2. (p < 0.0001, d = 1.2) 

Blum, 2006: Main concerns were emotional support (80%) and meeting and negotiating the need of self and family (72%)

Seers, 2009: 1. Psychological and emotional 2. Concerns about well-being; Significant improvement of concerns 1. (p < 0.0005, d =1.45), 2. (p < 0.0005, d = 1.23) and well- being (p < 0.0005, d = 0.49) after treatment”

DISCUSSION

As mentioned, consider expanding on the terms of counselling and “supportive psychotherapy” further, with the implication of the decision to include /exclude “supportive psychotherapy” in the search strategies discussed.

Reply: We have edited the whole manuscript according to the changes we have made resulting from your first comment. 

CONCLUSION

The starting sentence in the Conclusion of the manuscript does not match the primary concluding remark in the Abstract. Perhaps consider restructuring the sentences in the Conclusion section of the manuscript so that the lack of high-quality research is highlighted at the start of the section (e.g. bringing the sentence “The current state of research…is unsatisfactory” [line 264] forward and place the sentence on the suggested effectiveness of cancer counselling [line 258] later in the section).

Reply: Thank you for the suggestion. The conclusion was changed accordingly.

Sentences (line 260-263) justifying the significance of counselling services compared to psychotherapy might be best placed elsewhere in the manuscript (e.g. introduction or discussion) with elaboration on the similarities and differences between them.

Reply: Thank you for this suggestion. We have deleted the sentence as it did not comply with the changes we have made in the introduction and the rest of the manuscript.

Figure 1: Suggest adding caption.

Reply: Thank you for pointing this out. We checked the original manuscript and did add a caption to our figure in the text: “Fig 1. Flowchart: Study screening and selection.” However, we are uncertain whether we had to state the caption somewhere during the uploading process elsewhere. We will check with the PLOS ONE criteria to avoid any loss of information. 

Table 2: Overview of selected articles: 1. Consider adding in the quantification of significant improvement for particular study (e.g. >30% improvement) to allow for easier appreciation of the effect size. 2. Consider adding the frequency intervention as mentioned above

Reply: Thank you! 1. We have included the quantification of significant results in form of effect sizes in table 3 as suggested. 2. We have included the frequency of the interventions, where stated in the articles as mentioned above. 

Table 3: Instruments and key finding: There are some overlaps of contents between Table 2 and Table 3. Perhaps, the overlapping contents can be integrated?

Reply: Table 3 was included to provide additional and more detailed information on the instruments and key findings. For better comprehension of the tables and to avoid any overlaps, we have divided the content into overview and characteristics of the studies (table 2) and instruments and key findings (table 3).

S1 Table: Full search strategy. Elaborating on how the various search items are integrated in the final search would allow reproducibility. (e.g. Are all #1-15 joined with “OR”?). Consider adding in “supportive psychotherapy” in the search as suggested earlier.

Reply: Thank you for the remark. All search terms #1-15 where indeed joined with “OR”, which we have edited in S1 Table. We have also included the additional search with “supportive psychotherapy”. 

With revision, this study would likely be an important contribution to the current literature where evidence for counselling service in this context is lacking.

Reviewer #2: 

Thank you for the opportunity to review this study. The review seeks to examine the literature to determine the scope of studies examining cancer counselling services. The topic area is important and the manuscript, for the most part, is concise and well-constructed. Nevertheless, the language needs to be tightened throughout. There are several colloquialisms or poorly worded sentences in the manuscript. It could be greatly improved by some careful editing. Further suggestions for improving the manuscript are outlined below.

Intro

- Line 47-48: Perceived as distressing – is this for cancer patients? Please clarify.

Reply: This sentence is indeed unclear. Thank you. We have clarified the section as suggested: “[…] are perceived as distressing by patients with cancer.” (line 47)

- Could consider using person first language, i.e. ‘patients with cancer’ rather than ‘cancer patients’.

Reply: Thank you for this important suggestion. We have changed all phrasing concerned to first person language.

- The authors don’t address any previous reviews examining the area of psychosocial cancer care in the background section. There have certainly been reviews in this domain. It would be helpful if the authors addressed this and identified how their current study fits in and builds on this literature.

Reply: Thank you very much for the remark. We have edited the introduction as a result of the other reviewer extensively and as you suggested included a number of reviews on the topic: “The efficacy of psychosocial support for patients with cancer has been established for improving the quality of life, decreasing levels of distress and the risk for developing depression or anxiety disorders (14-18)” (line 65-67).

Methods

- Eligibility criteria: Was there any restriction on study types? Would be good to include study designs that were excluded. Further, a rationale behind why some of the services (psychotherapy, online etc.) were excluded would be helpful.

Reply: We did not have any criteria for the study type as a scoping review allows including different study designs and we aimed to provide a general overview of the literature in the field. We included a rational for our inclusion and exclusion criteria: 

“Additionally, we will focus solely on support delivered face-to-face since overviews of telephone and online support has been reported elsewhere (19-21)” (line 79-81).

“Only studies assessing support delivered face-to-face were considered because support delivered online or by telephone may also include patients in the in-patient setting, which does not comply with our eligibility criteria. Moreover, there are a number of reviews that have already summarized the literature on telephone and online support interventions. Therefore, articles that evaluated telephone or online support were excluded.“ (line 95-99)

- Data charting process and data items: Were discussions to resolve just between the two coding authors or were additional authors consulted? Please clarify.

Reply: All discrepancies were resolved among the two researchers. The section was edited for clarification: “Two independent reviewers conducted the search. […]. Differences regarding the final selections of articles were resolved by discussing the choices until a consensus was reached among the two researchers.“ (line 114-117). 

- Synthesis of results: Some of this information is not really needed here. Would consider removing the first two sentences and sentence four.

Reply: The sentences were deleted as suggested. Thank you. 

Results

- Characteristics: Would suggest trying be more explicit with all the outlined characteristics. For example, rather than ‘Most articles reported a sample size between xxx’, should state ‘XX articles reported a sample size <100, XX between 101-400 and XX >400’. Obviously with consideration to the sample size spread or informed by literature on appropriate sample size.

Reply: Thank you for the suggestion. We have adjusted the section as you suggested: 

“One article stated a sample size of < 100 participants (27). Eight articles reported a sample size between 101 – 500 participants (25, 26, 28-30, 32-34). Two articles included between 501 – 1000 participants (31, 36) and one study included 1930 participants (35).” (line 161-164).

- Table 2: Not sure I understand why the journal needs to be included in this table. I would suggest including the lead author, year and country all in one column (the first column). Remove journal and perhaps move the study design to the second or third column to contextualise the study. It is indicated that for the Key findings the P-value would be included in the table but none are reported and most don’t include proportions.

Reply: Thank you for this remark. We have edited the first column as you suggested and moved the study design to the second column. We have divided the content into overview and characteristics of the studies (table 2) and instruments and key findings (table 3). We have also added p-values and effect sizes to the key findings (table 3).

- Outcomes: I don’t think the way the outcomes are presented is the optimal way to present these results. Grouping by outcome measures implies that it is the measure that has impacted the findings rather than the intervention/perceptions of patients. I would suggest presenting the outcomes grouped by studies instead.

Reply: We agree with you and have edited the outcome section and reported the results grouped by study and study design. 

- There is no discussion of the methodological coding applied and how the scoring was applied to each study.

Reply: We included a table with specific information on the evaluation and coding of each study with the EPHPP Quality Assessment Tool (S3 Table. Quality Assessment (EPHPP)): “Two independent researcher assessed the quality of the studies. Where the ratings did not correspond, a third researcher evaluated the criteria.” (line 133-135).

Discussion

- Further to my comment in the introduction, I am not sure this is the only review on this topic. It would be good for the authors to consider reviews in this field to highlight the gap they are addressing.

Reply: Thank you again for the suggestion. As we have included reviews and strengthened the aim of this review, we were able to highlight the gap this review addressed: “To the authors’ knowledge, this is the first literature review focusing on studies that assessed psychosocial support services for patients with cancer implemented in out-patient health care.“ (line 211-213).

- Line 177 – remove ‘great’. Need to be cautious using terms like this in scientific writing.

Reply: We removed ‘great’ as suggested. We also corrected the whole article for colloquial language.

- Lines 188-191: Could you provide a bit more detail on the studies you refer to? Were these studies population level? Were they outside the scope of this review?

Reply: Thank you for your comment. The studies were outside of the review and one article assessed the prevalence. We edited the sentence to clarify our argument: “The finding, that across all studies women primarily seek help, is consistent with studies on gender differences in attitudes towards help-seeking and desires for psychosocial support, where women show a greater desire for and more positive attitudes towards support compared to men (44, 45).“ (line 230-233).

- Line 189: Most services evaluated in this review were suitable – not just ‘most services are suitable’. Current phrasing is too broad and can’t be stated with accuracy.

Reply: Thank you, the phrasing was changed: “Ten support services evaluated in this review except two, which targeted specifically breast cancer patients, were suitable to support patients with any cancer diagnosis, which is also in line with studies assessing multiple cancer support services (46).“ (line 234-236). 

- The third paragraph of the discussion is quite long and convoluted. It is difficult to follow the authors’ arguments in sections. I would recommend trying to refine this paragraph and talk about the points more explicitly.

Reply: Thank you for the advice. We have divided the paragraph in sub parts and extended the discussion of each point as you suggested. Please refer to line 245-286.

- Not sure I agree that the instrument variation indicates a lack of research. It would be unusual for there to be a tool specifically targeting a service and population group. Differing outcome measures across quality of life, depression, anxiety etc. is very common in psychosocial literature. Agree that there is scope to talk about standardised psychosocial instruments for oncology patients, but I think the way this is currently written could be misleading.

Reply: We agree with you, the phrasing of the section may be misleading. We have edited the section extensively. Please refer to line 245-249.

- Para 4 on methodology basically just talks about the coding tool used – there is no real discussion of quality here and implications for the field. This paragraph needs to be strengthened.

Reply: Thank you for the remark. We added a discussion of the quality and implication: ”From the included articles, nine were rated as “weak”. This is mainly due to the cross-sectional study design and the absence of standardized instruments. The study of Götze and colleagues (2016), which was rated as “strong” demonstrates that it is possible to conduct high-quality research in the field (28). The quality assessment suggests that research on the effectiveness of implemented support services in cancer needs to be improved.“ (line 293-298).

- The delineation between cancer counselling services and psychotherapy is not clear in the manuscript so it makes it hard to understand some of the final conclusions. Could consider adding some detail to this in the intro.

Reply: Yes, we agree! With the changes that we have made throughout the whole manuscript, we hope that the quality and clarity of the review has improved and the objective and conclusion are more in line.

---

## [Decision Letter · Decision Letter 1]

2 Feb 2021

PONE-D-20-17447R1

Evaluations of psychosocial cancer support services: A scoping review

PLOS ONE

Dear Dr. Lingens,

Thank you for submitting your manuscript to PLOS ONE. After careful consideration, we feel that it has merit but does not fully meet PLOS ONE’s publication criteria as it currently stands. Therefore, we invite you to submit a revised version of the manuscript that addresses the points raised during the review process.

We look forward to receiving your revised manuscript.

Kind regards,

Tim Luckett

Academic Editor

PLOS ONE

Reviewers' comments:

Reviewer's Responses to Questions

**Comments to the Author**

1. If the authors have adequately addressed your comments raised in a previous round of review and you feel that this manuscript is now acceptable for publication, you may indicate that here to bypass the “Comments to the Author” section, enter your conflict of interest statement in the “Confidential to Editor” section, and submit your "Accept" recommendation.

Reviewer #1: All comments have been addressed

Reviewer #2: (No Response)

2. Is the manuscript technically sound, and do the data support the conclusions?

Reviewer #1: (No Response)

Reviewer #2: Partly

3. Has the statistical analysis been performed appropriately and rigorously? 

Reviewer #1: (No Response)

Reviewer #2: Yes

4. Have the authors made all data underlying the findings in their manuscript fully available?

Reviewer #1: (No Response)

Reviewer #2: Yes

5. Is the manuscript presented in an intelligible fashion and written in standard English?

Reviewer #1: (No Response)

Reviewer #2: Yes

6. Review Comments to the Author

Reviewer #1: (No Response)

Reviewer #2: The authors have done a substantial amount of work to address previous reviewer comments, much of which has helped improve the manuscript. However, some of the additions appear to be rushed and not well thought-through. The lack of editing and refinement in some of these sections (particularly in the introduction) make it hard to follow. Based on the updated manuscript, I have outlined some further modifications that I think may help improve this work.

Abstract

- Rather than ‘most studies’ and ‘Few studies’, please list the number for each.

- Please change ‘the support services evaluated’ to ‘the evaluated support services’.

- Given the low number of studies and lack of RCTs, I think the results statement needs to be more tentative. For instance, ‘while the included studies indicate some improvements to well-being for patients with cancer, the low number and lack of high quality of studies indicate these findings should be interpreted with caution.’

- Suggest removing the last statement. Don’t only need more high quality studies for this, but also need high quality studies to determine that the interventions are effective. This statement is a jump in logic.

Introduction

- First sentence of paragraph 2 could be refined, for example – ‘A cancer diagnosis can have a substantial impact on an individual’s mental health and wellbeing, which may require diverse psychological and social support’.

- Replace ‘form of delivery’ with ‘delivery format’

- Not sure I understand the relevance of the sentence on Pg 5, lines 75-76 (professionals from different fields etc.).

- The additional sections of the introduction, specifically the second and third paragraphs are difficult to follow. The paragraphs lack refinement and the logic is not clear. The second paragraph focuses on the flexibility of services though the relevance of this information is not established. There appears to be a vague link in the third paragraph where there is mention of outpatient settings having differing content etc. but again the link to the current paper isn’t clear. I think these sections would benefit from greater thought and refinement.

- In the last paragraph of the introduction (page 6) – could the authors remove ‘we’ from all sentences to make the sentences objective. E.g. ‘For this review psychosocial cancer support services are defined as…’

- Similar to above, the last sentence requires refinement – ‘A scoping review will provide an overview of the available literature to guide future evaluations of this evidence base’…or similar. Indicating why scoping reviews are useful and specifically why one is needed on this topic would be preferential to the current justifications.

Methods

- Please provide the references to the reviews that have already been conducted on page 7, lines 126-127.

- The justification that previous studies have examined telephone and online interventions is sufficient for exclusion and more compelling than the justification re the possible inclusion of inpatients.

- Page 7, lines 128-130: This sentence is difficult to follow. Indicating that supportive psychotherapy was excluded from your exclusion criteria for psychotherapy is a confusing way to present this information.

- Page 8, line 151 – remove ‘journal’ from the ‘information on the article’ list.

Results

- Page 19, line 233 – think there is a typo here. Perhaps ‘measured’ should be ‘measures’?

- Page 26, lines 240-244 – long, convoluted sentence. Difficult to follow.

- Page 26, line 244-245 – Needs revising, perhaps ‘All X pre and post studies using the MYCaW demonstrated improvements in general well-being’.

- Rather than ‘one study’ could state the authors name followed by et al.

- Page 26, line 247 – consider replacing ‘revealed’ with ‘demonstrated’ or ‘found’.

- Page 26, line 252 – ‘All articles’ appears to refer to a subset of articles (perhaps those that examined concerns?). Please clarify.

Discussion

- Page 27, lines 269-271 – I’m not sure I understand why the heterogeneity of the psychosocial support would result in a lack of RCTs? Also when you say it is difficult to have a control group when implementing an intervention in health care, is this attempting to suggest that withholding support would be unethical in practice? Please clarify.

- Page 27, line 276-279 – The review does not examine actual support services, it examined evaluation of services (i.e. research studies). Stating that support services are more prevalent in certain countries and approached by more women than men is not a finding of this research. It should be stated that the evaluations were conducted primarily in Europe and North America and that a higher number of women than men participated in these studies etc.

- Similar to above – this review didn’t examine whether women were more likely to seek help but rather observed a higher number of women participating in psychosocial support evaluations. This may reflect a greater willingness of women to seek support – but this should be tentative and not presented as a direct finding since it may also be due to study recruitment strategies (e.g. targeting breast cancer) or women being more likely to participate in research studies.

- Page 28, lines 289-292 – this sentence needs to be edited for clarity.

- Page 29, lines 321-325 – These statements are a bit unclear. Are the authors saying the measure examine concerns followed by well-being, which may bias the participant responses to rate their well-being higher?

- Page 30, lines 359-361 – as with the introduction, please remove reference to ‘we decided’. Rather provide the scientific rationale behind providing methodological coding. E.g. ‘While a scoping review does not require quality assessment, this was included for the current review to provide greater insight on the limited number of studies.’

- Information on the EPHPP (i.e. what it assessed and how the rating works) should be included in methods not discussion.

- I would also suggest talking about the methodological findings prior to the study findings in the discussion, as this sets up the reader to understand the quality of the studies being presented.

- Page 31, line 386 – interpreting the origin of the research does not need to be done with caution, the origin of the research is clearly stated in the research articles. Suggest removing this statement.

- Replace ‘gives rise to’ with ‘highlights’.

- Conclusion – as per comment about the abstract, there is a jump in logic for improving evidence would increase funding. Rather clear evidence indicating that psychosocial support services improve outcomes for patients with cancer may result in increased funding.

7. PLOS authors have the option to publish the peer review history of their article (what does this mean?). If published, this will include your full peer review and any attached files.

Reviewer #1: **Yes: **Wei Lee

Reviewer #2: No

---

## [Author Response · Author response to Decision Letter 1]

3 Mar 2021

Reviewer #1: Thank you again for you previous comments and for accepting the manuscript!

Reviewer #2: The authors have done a substantial amount of work to address previous reviewer comments, much of which has helped improve the manuscript. However, some of the additions appear to be rushed and not well thought-through. The lack of editing and refinement in some of these sections (particularly in the introduction) make it hard to follow. Based on the updated manuscript, I have outlined some further modifications that I think may help improve this work.

Abstract

- Rather than ‘most studies’ and ‘Few studies’, please list the number for each.

Reply: Thank you. We have made the changes as you suggested. 

- Please change ‘the support services evaluated’ to ‘the evaluated support services’.

Reply: We agree and have adjusted the wording.

- Given the low number of studies and lack of RCTs, I think the results statement needs to be more tentative. For instance, ‘while the included studies indicate some improvements to well-being for patients with cancer, the low number and lack of high quality of studies indicate these findings should be interpreted with caution.’

Reply: That is a good suggestion, so we have included your formulation in the abstract. 

- Suggest removing the last statement. Don’t only need more high quality studies for this, but also need high quality studies to determine that the interventions are effective. This statement is a jump in logic.

Reply: We have deleted the last statement and strengthened the conclusion as proposed by you.

Introduction

- First sentence of paragraph 2 could be refined, for example – ‘A cancer diagnosis can have a substantial impact on an individual’s mental health and wellbeing, which may require diverse psychological and social support’.

Reply: We believe your formulation is more precise and have edited the first sentence accordingly.

- Replace ‘form of delivery’ with ‘delivery format’

Reply: Agree!

- Not sure I understand the relevance of the sentence on Pg 5, lines 75-76 (professionals from different fields etc.).

Reply: We have deleted the sentence to improve the clarity of the paragraph.

- The additional sections of the introduction, specifically the second and third paragraphs are difficult to follow. The paragraphs lack refinement and the logic is not clear. The second paragraph focuses on the flexibility of services though the relevance of this information is not established. There appears to be a vague link in the third paragraph where there is mention of outpatient settings having differing content etc. but again the link to the current paper isn’t clear. I think these sections would benefit from greater thought and refinement.

Reply: Thank you for your remark. We agree that the point we are trying to make, could be clearer. Thus, we have combined paragraph 2 and 3 and strengthened the line of argumentation here:

“There is some indication of the efficacy of psychosocial support for patients with cancer in improving the quality of life, decreasing levels of distress and the risk for developing depression or anxiety disorders (11-15). The interventions tested for efficacy and summarized in the reviews are often manualized or tested in standardized settings for the purpose of the studies. However, the interventions often need to be adapted according to the diverse needs of patients with cancer when implemented in the health care setting (16-18). Hence, implemented psychosocial interventions may be flexible in duration, contents, delivery format (e.g., inpatient or outpatient setting and online, face-to-face or telephone) and may be administered by different health care staff (e.g., psychologists, nurses, trained volunteers) (8, 16). The efficacy of psychosocial interventions investigated in a scientific study may not be transferable to implemented psychosocial support since some aspects may change with different types, contexts and set-ups of the support services. Therefore evaluations of implemented psychosocial support in out-patient health care are important to assess the effectiveness considering their ecological validity and quality standards. The current literature and reviews on the topic reveal a scarcity of evaluations of implemented psychosocial support.“ (line 58-72)

- In the last paragraph of the introduction (page 6) – could the authors remove ‘we’ from all sentences to make the sentences objective. E.g. ‘For this review psychosocial cancer support services are defined as…’

Reply: We have deleted ‘we’ from all sentences. 

- Similar to above, the last sentence requires refinement – ‘A scoping review will provide an overview of the available literature to guide future evaluations of this evidence base’…or similar. Indicating why scoping reviews are useful and specifically why one is needed on this topic would be preferential to the current justifications.

Reply: We have made the last sentence more precise. We have not added any further justifications for choosing a scoping review, as the mere definition of scoping reviews is to provide a first overview of the available literature, which we have stated.

Methods

- Please provide the references to the reviews that have already been conducted on page 7, lines 126-127.

Reply: We have included the references.

- The justification that previous studies have examined telephone and online interventions is sufficient for exclusion and more compelling than the justification re the possible inclusion of inpatients.

Reply: We agree and have edited the section accordingly.

- Page 7, lines 128-130: This sentence is difficult to follow. Indicating that supportive psychotherapy was excluded from your exclusion criteria for psychotherapy is a confusing way to present this information.

Reply: We understand how it could be confusing. Therefore, we deleted ‘supportive psychotherapy’ from the section and mentioned it in the inclusion criteria section. 

- Page 8, line 151 – remove ‘journal’ from the ‘information on the article’ list.

Reply: Thank you for noticing. We have removed ‘journal’.

Results

- Page 19, line 233 – think there is a typo here. Perhaps ‘measured’ should be ‘measures’?

Reply: Yes, you are right. It is supposed to be ‘measures’.

- Page 26, lines 240-244 – long, convoluted sentence. Difficult to follow.

Reply: We have shortened the sentence to eliminate any confusion: “The results of the RCT showed that physical well-being improved after the support compared to the control group but not psychological and social well-being”

- Page 26, line 244-245 – Needs revising, perhaps ‘All X pre and post studies using the MYCaW demonstrated improvements in general well-being’.

Reply: Thank you. The sentence now reads as follows: “All four pre- and post- studies using the MYCaW demonstrated improvement in general well-being”

- Rather than ‘one study’ could state the authors name followed by et al.

Reply: Thank you, we have included the name. 

- Page 26, line 247 – consider replacing ‘revealed’ with ‘demonstrated’ or ‘found’.

Reply: We have replaced it with ‘found’.

- Page 26, line 252 – ‘All articles’ appears to refer to a subset of articles (perhaps those that examined concerns?). Please clarify.

Reply: Yes, we’ve meant all articles that examined concerns and have added this information.

Discussion

- Page 27, lines 269-271 – I’m not sure I understand why the heterogeneity of the psychosocial support would result in a lack of RCTs? Also when you say it is difficult to have a control group when implementing an intervention in health care, is this attempting to suggest that withholding support would be unethical in practice? Please clarify.

Reply: You are right. The section lacks clarity. We have strengthened the section and explanation: “The lack of studies with RCT designs may be due to the lack of standardized interventions offered in psychosocial support, which may hamper comparability. Another reason may be the difficulty of integrating a control group when assessing an intervention already implemented in health care, since it would be unethical to deny support for patients in need. One solution could be waitlist-control trials.” (line 213-217)

- Page 27, line 276-279 – The review does not examine actual support services, it examined evaluation of services (i.e. research studies). Stating that support services are more prevalent in certain countries and approached by more women than men is not a finding of this research. It should be stated that the evaluations were conducted primarily in Europe and North America and that a higher number of women than men participated in these studies etc.

Reply: You are correct, we examine evaluations of support services. However, we believe that it is valuable to also discuss the findings of those evaluations. Nevertheless we agree, that we need to clarify that these are not direct findings and that there are several ways to interpret the results. The origin of the studies may be due to a stronger focus of research in the field for those countries, which again may be i.e., due to resources available. On the contrary, it may also be the case that psychosocial support services are more common in those countries and thus there may be more need for research. We have edited this section in the discussion so it becomes clearer, that there are different interpretation to the findings.

- Similar to above – this review didn’t examine whether women were more likely to seek help but rather observed a higher number of women participating in psychosocial support evaluations. This may reflect a greater willingness of women to seek support – but this should be tentative and not presented as a direct finding since it may also be due to study recruitment strategies (e.g. targeting breast cancer) or women being more likely to participate in research studies.

Reply: Similar as above – we agree, it should not be presented as direct finding. It may be due to the study design or approach that across all studies more women were included because women are more likely to participate in studies. Since we assume that most studies considered the external validity and generalizable of their sample, we believe that it can be valuable to discuss the results and trends of the studies evaluating support services. 

- Page 28, lines 289-292 – this sentence needs to be edited for clarity.

Reply: We have changed the sentence to improve the clarity: “Two support services targeted specifically breast cancer patients and ten support services offered support for patients with any cancer diagnosis, which is also in line with the findings of a study assessing the characteristics of patients in multiple cancer support centers” (line 245-247).

- Page 29, lines 321-325 – These statements are a bit unclear. Are the authors saying the measure examine concerns followed by well-being, which may bias the participant responses to rate their well-being higher?

Reply: Yes, this is what we indented to express with the sentences. We clarified the section: “General and psychological well-being measured by the MYCaW and FACIT-SpEx resulted in clinically significant improvement (26, 29-31). The improvement of well-being when measured by the MYCaW may be due to the proximity of its assessment with that of the concerns (Concerns are examined followed by well-being). The patient may connect the successful resolution of the concerns with an improvement of well-being. Further research is needed to study the improvement of well-being when assessed in combination with concerns in the setting of psychosocial support.” (line 265-272)

- Page 30, lines 359-361 – as with the introduction, please remove reference to ‘we decided’. Rather provide the scientific rationale behind providing methodological coding. E.g. ‘While a scoping review does not require quality assessment, this was included for the current review to provide greater insight on the limited number of studies.’

Reply: Yes, thank you! We like your suggestion and have exchanged it.

- Information on the EPHPP (i.e. what it assessed and how the rating works) should be included in methods not discussion.

Reply: We agree with you that the methodological part of the discussion needs to be moved to the method and the discussion of the EPHPP should be at the beginning of the discussion. (See line 128-135 and line 222-228)

- I would also suggest talking about the methodological findings prior to the study findings in the discussion, as this sets up the reader to understand the quality of the studies being presented.

Reply: See our comment above. 

- Page 31, line 386 – interpreting the origin of the research does not need to be done with caution, the origin of the research is clearly stated in the research articles. Suggest removing this statement.

Reply: Yes we agree, that sentence is irritating so we have deleted it. 

- Replace ‘gives rise to’ with ‘highlights’.

Reply: We thank you for the suggestion and have changed it. 

- Conclusion – as per comment about the abstract, there is a jump in logic for improving evidence would increase funding. Rather clear evidence indicating that psychosocial support services improve outcomes for patients with cancer may result in increased funding.

Reply: We agree that there is a jump in logic and adjusted the conclusion accordingly.

---

## [Decision Letter · Decision Letter 2]

16 Mar 2021

PONE-D-20-17447R2

Evaluations of psychosocial cancer support services: A scoping review

PLOS ONE

Dear Dr. Lingens,

Thank you for submitting your manuscript to PLOS ONE. After careful consideration, we feel that it has merit but does not fully meet PLOS ONE’s publication criteria as it currently stands. Therefore, we invite you to submit a revised version of the manuscript that addresses the points raised during the review process.

We look forward to receiving your revised manuscript.

Kind regards,

Tim Luckett

Academic Editor

PLOS ONE

Journal Requirements:

Additional Editor Comments (if provided):

Please insert headings into the Abstract including Background, Methods, Results and Conclusions.

In the manuscript, please make changes as follows:

Methods

- Insert the heading Quality Assessment or Risk of Bias before discussing the EPHPP;

- Further develop the synthesis section to explain why a meta-analysis was not considered possible where the same measure was used in multiple studies (presumably related to the statement in the Discussion that "the heterogeneity of the services and their offers makes it difficult to interpret and compare the outcomes across the selected studies").

- Please also include a brief description of the narrative approach taken to comparing results across studies.

Results

- I think Table 3 can be deleted and the final column moved to Table 2. Table 2 already includes details of measures, and the validity/reliability columns are themselves neither valid nor reliable given that these are continuous rather than dichotomous variables. To make more space in Table 2, I recommend moving the quality rating to a new table where rating of each characteristic is reported as well as the overall grade.

Discussion

Please remove the sub-heading 'summary of evidence'.

I think the author's have made assumptions regarding the nature of scoping vs systematic reviews that are not justified given the various ways in which the former term is used. Please remove the following sentences:

"As the small number of suitable articles suggests, the scoping review format was a reasonable choice"

"While a scoping review does not require a quality assessment, this was included for the 222 current review to provide greater insight on the limited number of studies" (changing the sentence that follows to clarify it is referring to quality rating).

Reviewers' comments:

Reviewer's Responses to Questions

**Comments to the Author**

1. If the authors have adequately addressed your comments raised in a previous round of review and you feel that this manuscript is now acceptable for publication, you may indicate that here to bypass the “Comments to the Author” section, enter your conflict of interest statement in the “Confidential to Editor” section, and submit your "Accept" recommendation.

Reviewer #2: All comments have been addressed

2. Is the manuscript technically sound, and do the data support the conclusions?

Reviewer #2: Yes

3. Has the statistical analysis been performed appropriately and rigorously? 

Reviewer #2: Yes

4. Have the authors made all data underlying the findings in their manuscript fully available?

Reviewer #2: Yes

5. Is the manuscript presented in an intelligible fashion and written in standard English?

Reviewer #2: Yes

6. Review Comments to the Author

Reviewer #2: Thank you for the updates to the manuscript. I suggest the minor editorial updates listed below and then accept the manuscript. Well done.

Pg 4. Ln 63-64. Please modify to: The interventions tested for efficacy and summarized in the reviews are largely controlled and standardized to reduce the likelihood of experimental bias.

Pg 4. Ln 69-70. Please modify to: The efficacy of psychosocial interventions investigated in an experimental study may not be transferable when implemented to real-world psychosocial support services, due to the different settings and contexts of the support services.

Pg 4. Ln 72-73. Please modify to: …assess their effectiveness with consideration to their ecological validity and quality.

Pg 4. Ln 73-74. Please modify to: There is currently no evaluation of implemented psychosocial support within out-patient cancer settings.

Pg 5. Ln 97-98. Please modify to: This review focuses on face to-face delivered support since overviews of telephone and online support has been reported elsewhere (19-21).

Pg 5. Ln 117-118. Please modify to: Studies evaluating psychotherapy aimed at treating mental disorders were excluded.

Pg 7. Ln 155: Please delete “It is important to note that”.

Pg 17. Ln 212: Please change “support” to “supportive intervention”. Further, please change latter part of sentence to: “but psychological and social well-being did not”.

Pg 18. Ln 242-243” Please move sentence to the end of the paragraph so it reads: “One possible study design to overcome this concern could involve using wait-list controls”.

Pg 18. Ln 249: Replace “mainly” with “largely”

Pg 19. Ln 265-267: Please modify to: Greater participation of women than men in the evaluated studies may be due to increased willingness of women to participate in research studies.

7. PLOS authors have the option to publish the peer review history of their article (what does this mean?). If published, this will include your full peer review and any attached files.

Reviewer #2: No

---

## [Author Response · Author response to Decision Letter 2]

31 Mar 2021

Response to Reviewers

Reviewer #2: Thank you for the updates to the manuscript. I suggest the minor editorial updates listed below and then accept the manuscript. Well done.

Reply: Thank you very much! We have incorporated the changes as you suggested below.

Pg 4. Ln 63-64. Please modify to: The interventions tested for efficacy and summarized in the reviews are largely controlled and standardized to reduce the likelihood of experimental bias.

Pg 4. Ln 69-70. Please modify to: The efficacy of psychosocial interventions investigated in an experimental study may not be transferable when implemented to real-world psychosocial support services, due to the different settings and contexts of the support services.

Pg 4. Ln 72-73. Please modify to: …assess their effectiveness with consideration to their ecological validity and quality.

Pg 4. Ln 73-74. Please modify to: There is currently no evaluation of implemented psychosocial support within out-patient cancer settings.

Pg 5. Ln 97-98. Please modify to: This review focuses on face to-face delivered support since overviews of telephone and online support has been reported elsewhere (19-21).

Pg 5. Ln 117-118. Please modify to: Studies evaluating psychotherapy aimed at treating mental disorders were excluded.

Pg 7. Ln 155: Please delete “It is important to note that”.

Pg 17. Ln 212: Please change “support” to “supportive intervention”. Further, please change latter part of sentence to: “but psychological and social well-being did not”.

Pg 18. Ln 242-243” Please move sentence to the end of the paragraph so it reads: “One possible study design to overcome this concern could involve using wait-list controls”.

Pg 18. Ln 249: Replace “mainly” with “largely”

Pg 19. Ln 265-267: Please modify to: Greater participation of women than men in the evaluated studies may be due to increased willingness of women to participate in research studies.

Responses to Editor

Additional Editor Comments (if provided):

Please insert headings into the Abstract including Background, Methods, Results and Conclusions.

Reply: We included the headings as suggested. 

In the manuscript, please make changes as follows:

Methods

- Insert the heading Quality Assessment or Risk of Bias before discussing the EPHPP;

Reply: We have inserted the heading quality assessment. 

- Further develop the synthesis section to explain why a meta-analysis was not considered possible where the same measure was used in multiple studies (presumably related to the statement in the Discussion that "the heterogeneity of the services and their offers makes it difficult to interpret and compare the outcomes across the selected studies").

- Please also include a brief description of the narrative approach taken to comparing results across studies.

Reply: Thank you for the request. We have included an explanation as suggested for the choice against a meta-analysis and our narrative approach to comparing the results: 

“For articles that used the same instruments a meta-analysis would be the preferable choice. However, due to the heterogeneity of the setting and context of psychosocial support services, comparability is not guaranteed. Hence, a scoping review was the reasonable alternative.” (line 136-138)

“Furthermore, since different instruments were used for the evaluation of services, the results were compared across studies within similar theoretical concepts e.g. satisfaction with the service. The global scores and details of the quality assessment are reported in Table 1.” 

(line 142-145)

Results

- I think Table 3 can be deleted and the final column moved to Table 2. Table 2 already includes details of measures, and the validity/reliability columns are themselves neither valid nor reliable given that these are continuous rather than dichotomous variables. To make more space in Table 2, I recommend moving the quality rating to a new table where rating of each characteristic is reported as well as the overall grade.

Reply: Thank you, we have deleted table 3 and added the last column to table 2. We included the quality rating (previously in the supporting information S3 Table) as a new table (Table 1) in the manuscript.

Discussion

Please remove the sub-heading 'summary of evidence'.

Reply: We have removed the subheading.

I think the author's have made assumptions regarding the nature of scoping vs systematic reviews that are not justified given the various ways in which the former term is used. Please remove the following sentences:

"As the small number of suitable articles suggests, the scoping review format was a reasonable choice"

"While a scoping review does not require a quality assessment, this was included for the 222 current review to provide greater insight on the limited number of studies" (changing the sentence that follows to clarify it is referring to quality rating).

Reply: We have deleted the sentences as suggested.

Journal Requirements:

Reply: The retracted article has been removed:

Gordon VM, Madhotra R, Steer P. A Single Centre Experience of Straight to Test Suspected Ugi Cancer Service. Gut. 2011;60.

Changes that have been made through-out the revision process to the reference list are listed below:

Removed:

Cousson-Gelie F, Bruchon-Schweitzer M, Atzeni T, Houede N. Evaluation of a Psychosocial Intervention on Social Support, Perceived Control, Coping Strategies, Emotional Distress, and Quality of Life of Breast Cancer Patients. Psychol Rep. 2011;108(3):923-42.

Boltong A, Ledwick M, Babb K, Sutton C, Ugalde A. Exploring the rationale, experience and impact of using Cancer Information and Support (CIS) services: an international qualitative study. Support Care Cancer. 2017;25(4):1221-8.

Öhlén J, Holm A-K, Karlsson B, Ahlberg K. Evaluation of a counselling service in psychosocial cancer care. Eur J Oncol Nurs. 2005;9(1):64-73.

Jevne RF. Looking back to look ahead: A retrospective study of referrals to a cancer counselling service. International Journal for the Advancement of Counselling. 1990;13(1):61-72.

Furzer BJ, Wright KE, Petterson AS, Wallman KE, Ackland TR, Joske DJ. Characteristics and quality of life of patients presenting to cancer support centres: patient rated outcomes and use of complementary therapies. BMC Altern Med. 2013;13:169.

Included:

Holland JC, Andersen B, Breitbart WS, Compas B, Dudley MM, Fleishman S, et al. Distress Management Clinical Practice Guidelines in Oncology (TM). J Natl Compr Canc Ne. 2010;8(4):448-85.

Morris SB. Estimating effect sizes from pretest-posttest-control group designs. Organ Res Methods. 2008;11(2):364-86.

Lebow J. Overview of psychotherapies UpToDate Inc.: Waltham, MA, USA.2019 [Available from: http://www.uptodate.com.

Page AE, Adler NE. Cancer care for the whole patient: Meeting psychosocial health needs: National Academies Press; 2008.

Watson M, Kissane DW. Handbook of Psychotherapy in Cancer Care. 1 ed: John Wiley & Sons, Inc.; 2011.

Gabriel I, Creedy D, Coyne E. A systematic review of psychosocial interventions to improve quality of life of people with cancer and their family caregivers. Nurs Open. 2020;7(5):1299-312.

Galway K, Black A, Cantwell M, Cardwell CR, Mills M, Donnelly M. Psychosocial interventions to improve quality of life and emotional wellbeing for recently diagnosed cancer patients. Cochrane Db Syst Rev. 2012(11).

Salsman JM, Pustejovsky JE, Schueller SM, Hernandez R, Berendsen M, McLouth LES, et al. Psychosocial interventions for cancer survivors: A meta-analysis of effects on positive affect. J Cancer Surviv. 2019;13(6):943-55.

Teo I, Krishnan A, Lee GL. Psychosocial interventions for advanced cancer patients: A systematic review. Psycho-Oncology. 2019;28(7):1394-407.

Vartolomei L, Shariat SF, Vartolomei MD. Psychotherapeutic Interventions Targeting Prostate Cancer Patients: A Systematic Review of the Literature. European urology oncology. 2018;1(4):283-91.

Lederberg MS, Holland JC. Supportive psychotherapy in cancer care: An essential ingredient of all therapy. Handbook of psychotherapy in cancer care. 2011;1.

Boltong A, Ledwick M, Babb K, Sutton C, Ugalde A. Exploring the rationale, experience and impact of using Cancer Information and Support (CIS) services: an international qualitative study. Support Care Cancer. 2017;25(4):1221-8.

Hong Y, Pena-Purcell NC, Ory MG. Outcomes of online support and resources for cancer survivors: A systematic literature review. Patient Educ Couns. 2012;86(3):288-96.

Okuyama S, Jones W, Ricklefs C, Tran ZV. Psychosocial telephone interventions for patients with cancer and survivors: a systematic review. Psycho-Oncology. 2015;24(8):857-70.

Munn Z, Peters MDJ, Stern C, Tufanaru C, McArthur A, Aromataris E. Systematic review or scoping review? Guidance for authors when choosing between a systematic or scoping review approach. Bmc Med Res Methodol. 2018;18.

Baker BS, Hoffman CJ, Fenlon D. What can a third sector organisation provide for people with breast cancer that public health services cannot? Developing support services in response to service evaluation. Eur J Integr Med. 2019;30:9.

Grassi L, Fujisawa D, Odyio P, Asuzu C, Ashley L, Bultz B, et al. Disparities in psychosocial cancer care: a report from the International Federation of Psycho-oncology Societies. Psycho-Oncology. 2016;25(10):1127-36.

Mackenzie CS, Gekoski WL, Knox VJ. Age, gender, and the underutilization of mental health services: The influence of help-seeking attitudes. Aging & Mental Health. 2006;10(6):574-82.

Merckaert I, Libert Y, Messin S, Milani M, Slachmuylder JL, Razavi D. Cancer patients' desire for psychological support: prevalence and implications for screening patients' psychological needs. Psychooncology. 2010;19(2):141-9.

Gessler S, Low J, Daniells E, Williams R, Brough V, Tookman A, et al. Screening for distress in cancer patients: is the distress thermometer a valid measure in the UK and does it measure change over time? A prospective validation study. Psycho-Oncology. 2008;17(6):538-47.

Doherty M, Miller-Sonet E, Gardner D, Epstein I. Exploring the role of psychosocial care in value-based oncology: Results from a survey of 3000 cancer patients and survivors. J Psychosoc Oncol. 2019;37(4):441-55

---

## [Editor Report · Decision Letter 3]

8 Apr 2021

PONE-D-20-17447R3

Evaluations of psychosocial cancer support services: A scoping review

PLOS ONE

Dear Dr. Lingens,

Thank you for submitting your manuscript to PLOS ONE. After careful consideration, we feel that it has merit but does not fully meet PLOS ONE’s publication criteria as it currently stands. Therefore, we invite you to submit a revised version of the manuscript that addresses the points raised during the review process.

Specifically, please replace "scoping review" with "narrative synthesis" in the following sentence: "Hence, a scoping review was the reasonable alternative".

We look forward to receiving your revised manuscript.

Kind regards,

Tim Luckett

Academic Editor

PLOS ONE

---

## [Author Response · Author response to Decision Letter 3]

9 Apr 2021

Academic editor:

Specifically, please replace "scoping review" with "narrative synthesis" in the following sentence: "Hence, a scoping review was the reasonable alternative

Reply: We edited the sentence as suggested.

---

## [Editor Report · Decision Letter 4]

21 Apr 2021

Evaluations of psychosocial cancer support services: A scoping review

PONE-D-20-17447R4

Dear Dr. Lingens,

We’re pleased to inform you that your manuscript has been judged scientifically suitable for publication and will be formally accepted for publication once it meets all outstanding technical requirements.

Kind regards,

Tim Luckett

Academic Editor

PLOS ONE

---

## [Editor Report · Acceptance letter]

23 Apr 2021

PONE-D-20-17447R4 

Evaluations of psychosocial cancer support services:A scoping review 

Dear Dr. Lingens:

I'm pleased to inform you that your manuscript has been deemed suitable for publication in PLOS ONE. Congratulations! Your manuscript is now with our production department. 

Kind regards, 

on behalf of

Dr. Tim Luckett 

Academic Editor

PLOS ONE